# Flexible Micro-Battery for Powering Smart Contact Lens

**DOI:** 10.3390/s19092062

**Published:** 2019-05-03

**Authors:** Mohamed Nasreldin, Roger Delattre, Marc Ramuz, Cyril Lahuec, Thierry Djenizian, Jean-Louis de Bougrenet de la Tocnaye

**Affiliations:** 1Mines Saint-Etienne, Center of Microelectronics in Provence, Department of Flexible Electronics, F-13541 Gardanne, France; mohamed.nasreldin@emse.fr (M.N.); roger.delattre@emse.fr (R.D.); ramuz@emse.fr (M.R.); thierry.djenizian@emse.fr (T.D.); 2IMT Atlantique, Optics & Electronics Departments, CS 83818, F-29238 Brest CEDEX 3, France; cyril.lahuec@imt-atlantique.fr

**Keywords:** smart contact lens, flexible micro battery, energy harvesting, ASIC

## Abstract

In this paper, we demonstrate the first attempt of encapsulating a flexible micro battery into a contact lens to implement an eye-tracker. The paper discusses how to scale the battery to power various circuits embedded in the contact lens, such as ASIC, photodiodes, etc., as well as how to combine the battery with external harvested energy sources. The fabricated ring battery has a surface area of 0.75 cm^2^ yielding an areal capacity of 43 µAh·cm^−2^ at 20C. Based on simulated 0.35-µm CMOS ASIC power consumption, this value is large enough to allow powering the ASIC for 3 minutes. The functioning of the micro battery is demonstrated by powering an orange LED.

## 1. Introduction

Several functions have already been successfully implemented on wireless contact lenses (CL) to provide various biosensors [1,2], most of them dealing with intraocular pressure (IoP) gauges [3]. One of the main limitations in the development of more advanced functions embedded on a contact lens, such as eye-tracking [4] or refractive correction [5], is related to the energy supply necessary for data processing and communication at the CL level. In general, this issue is solved by energy harvesting techniques [6], for which many solutions have been proposed on a contact lens (e.g., optical [7], mechanical [8], chemical [9], and radio frequency [10]). However, if energy harvesting should be considered in most of the cases, the need for buffering energy is mandatory when the harvested energy does not exceed the energy required to power the sensor. This arises whenever the sensor implements complex tasks and transmits data, by means of an RF link, for instance [2].

It is critical to know if the CL benefits from harvesting or if the battery is the only source of power (disposable contact lenses). Furthermore, is the external harvester used to partially (periodically) reload the battery, in parallel to powering continuously the CL circuits, or not? Such a concern directly impacts what is expected from the battery. Even if energy harvesting is crucial for long-term operations of a smart CL, a battery should be combined to the harvester, the power of being both designed and scaled according to the consumption needs, which depend upon the operations carried out on the CL. For instance, for the IoP described in Reference [3], having a power consumption of a few nW, the IoP gauge does not operate continuously, but in a raster mode. Furthermore, the main power consumption is not the gauge but the data transmission. Hence, the complexity of computing operations embedded in the CL as well as the monitoring regime (raster or continuous) have a direct impact on the harvester and battery complementary dimensioning.

In this paper, we present for the first time the integration of a new flexible Li-ion micro battery in a CL. The smart CL application considered in this work is an eye-tracking system for which complex calculations need to be performed into the CL to extract, for instance, the gaze direction [4].

The paper is organized as follows. In the first part, we describe the battery principle and the requirements the battery should meet before its encapsulation into the CL. In a second part, we measure the battery performance, and its proper functioning illustrated by powering an external visible LED. In the last part, we suggest some future extends and estimate the expected storage capability achievable using this technology (lifetime, maximum current provided, transparency, etc.).

## 2. System Power Needs and Battery Requirements

The battery characteristics (e.g., energy and power densities, rate capability, cyclability, lifetime, etc.) depend on operations carried out into the CL. First, the silicon chip that needs to be powered is described. Second, the different powering modes are specified. Finally, based on these results, the battery characteristics are drawn.

### 2.1. Eye-Tracker Case

The application targeted here is an eye-tracker embedded into a CL. The full system is detailed in Reference [4]. Basically, a centroid is computed by means of four infrared photo-sensors (IRP) encapsulated in the CL. The CL is illuminated by IR light from an external head gear (goggles, Head up Display etc.). Eye motions modify the light received by each IRP. The part of the system implemented into the CL (Figure 1) is briefly detailed, since it is battery powered. The application specific integrated circuit (ASIC) performing the gaze analysis is either powered by the battery or by the harvested RF power provided by the head gear. The ASIC has been designed and simulated for the AMS 0.35 µm-CMOS process. A detailed description of the ASIC is given in Reference [11]. To summarize, the design has taken into account the limited power that can be typically harvested (i.e., few mW). To reduce complexity and power consumption, the blocks composing the ASIC (e.g., processing unit, analog-to-digital converters, RF transmitter, etc.) have been designed to operate in the subthreshold region. This design technique allows using lower supply voltage than that typically required by the technology (here 1.2 V instead of 3.3 V for the AMS 0.35 µm-CMOS technology) along with ultra-low biasing currents, in the hundreds of nA range [12]. Both are leading to ultra-low power circuits. For instance, the processing block only consumes 0.043 mW, yielding a current of 36 µA. However, wirelessly transmitting data consumes a lot of energy. The simulated (SPICE) ASIC power consumption figures of Table 1 consider the minimum system functions required to perform the eye-tracking task and transmit the centroid coordinates, yielding an average power consumption of about 1 mW [11]. As shown in Table 1, altogether the other blocks consumes only one fourth of the total power consumption. However, some working phases could require higher power consumption for short time: start, calibration, etc. Other electronic functions are likely to be implemented (i.e., power regulation and management [13], calibration, power-on reset, etc.). Calibration might require an RF transceiver instead of a simple receiver, for the user to send orders (start/end sequences, etc.), resulting in an increase in the ASIC power consumption and the need for power buffering. Therefore, considering that using power management techniques, some blocks of the ASIC could be switched-off so that only necessary parts work to perform specific tasks, the system needs about 600 µA over 1.2 V to make the RF transmitter work. Hence, the RF transmitter power consumption dictates the power requirements and thus the dimensioning of the harvester and the battery.

### 2.2. Power Modes and RF Energy Harvesting

Energy harvesting can be achieved in different ways. For instance, biofuel harvesters in contact with lacrimal fluid use oxidation of glucose and/or ascorbate in tears. However, the harvested energy is small and depends on the anode/cathode surface, limited here. Typical values of 5–10 µW have been reported [9]. In addition, critical to the development of biofuel harvesters is the lifetime of such harvesters. In parallel, solar cells are difficult to encapsulate and harvested energy depends on the active surface as well. This is why RF harvesting is the most efficient and currently use for wireless contact lenses CL (see [14]). Let us consider, for instance, the harvesting regime described in Figure 2. The eye-tracker operates in a monitoring regime to measure fatigue and stress. In this case, the system operates as an interrogator (similar to IoP). Outside this time slot, the antenna is in a sleeping mode and the system is used to reload the battery, as shown in Figure 2. In the second case, during the eye-tracking regime, the direction of sight is measured continuously and sent to the external antenna at a minimum rate of 120 Hz. It corresponds to high power consumption due to continuous transmission of the date.

### 2.3. Battery Dimensioning and Requirements

The second aspect of the battery characteristics concerns the enabling technology. The fabrication of flexible/stretchable electrodes is a first challenge that can be overcome, by using for instance an approach we have recently reported [15]. We report the fabrication of lithium nickel manganese oxide (LNMO) micro pillars electrodes on Al serpentine interconnects that can be stretched up to 70% without structural damaging. We used a similar method as reported in Reference [16], where a mask-less technique relying on the laser patterning of metal tapes is used to make Au and Cu serpentine interconnects encapsulated in PDMS. The second challenge is to fabricate a high performance Li-ion micro-battery with a specific design, ensuring the light transmission, while fitting onto the surface offered by the CL meniscus (Figure 3) to be easily encapsulated, as detailed later on. Based on that, the battery should be dimensioned to output at least 600 µA on 1.2 V to power the RF block of the ASIC. The surface area available is also limited to the geometry of the CL. The center circle of the CL must remain free for the pupil. Hence, to maximize the battery surface area, it should have a ring shape and considering the scleral CL used, the available surface area is 0.75 cm^2^.

## 3. Principle of Encapsulation in a Scleral Contact Lens

We used contact lenses provided by the company LCS. Several families of contact lenses exist, mainly defined by the material and the dimensions of the lens. Rigid gas permeable contact lens (RGP) and scleral lenses are made with non-hydrophilic and oxygen permeable polymer materials. RGP lenses have a total diameter from 7.50 to 12.00 mm, whereas the total diameter for scleral lenses can be up to 16.50 mm. The contact lens type choice for the encapsulation determines the dimensions and the design of the flexible device (here the battery). For a larger surface of the CL families described hereinabove, the battery (and then the electronics) is encapsulated in a scleral contact lens. Their dimensions, the material they are made of and their stability on the eye make the scleral contact lens appropriate for prototype testing because they offer more options and space for the electronics dimensions.

In order to provide a final contact lens with a satisfactory design, the flexible battery substrate is encapsulated on a curved surface. An adaptable encapsulation process for each electronic function to be integrated and a custom scleral CL design has been developed by LCS in order to achieve the best optical properties in the pupil area, which should remain free of any electronics or substrate, Figure 4 in this current embodiment.

## 4. Micro-Battery Design and Fabrication

To be encapsulated into a contact lens, the batteries should meet several requirements in terms of design, size, thickness, discharge rate, and flexibility features. Even if several micro battery designs have been reported (e.g., Reference [17]), integration of a micro power source system fulfilling all these criteria in a contact lens has not been achieved so far. The innovative micro battery approach proposed here relies on two flexible substrates assembling consisting of polydimethylsiloxane (PDMS) supporting 1 cm^2^ surface area disk of serpentine electrodes separated by a gel polymer electrolyte (MA-PEG). First, metallic 30 µm thick aluminum foils were well cleaned using acetone, ethanol, and deionized water for 10 min each and then dried in a furnace at 100 °C for 30 min. Then, the aluminum foil was laminated onto a glass slide using a thermal release double sided 90 °C Nitto RevAlpha tape. PDMS (Sylgard 184) was spin coated on top of the aluminum foil and cured right after at 80 °C for 3 h. The Rev Alpha tape was released in a furnace at 100 °C for 15 min then the aluminum foil on top of PDMS membrane was laminated upside down on a glass substrate to cast, subsequently, the active material layer on top of it. Commercial Lithium Nickel Manganese Oxide (LNMO) and Lithium Titanate (LTO) powders typically serving as cathode and anode materials respectively for Li-ion batteries were purchased from MTI Corp, USA. The powders materials were mixed with carbon black (Super P) and polyvinylidene fluoride (PVDF) in the ratio of 90:5:5 and then were mixed with N-methyl-2-pyrrolidone (NMP) to obtain a paste that was doctor bladed on top of the aluminum foil. The electrode was dried under vacuum at 110 °C for 12 h to achieve LNMO and LTO electrodes thickness of 100 µm. Laser ablation technique was used to design the ring with serpentines. Laser treatments were carried out using the LPKF Protolaser S equipment with a radiation of 1064 nm, at 75 KHz, and a beam diameter of 25 µm. A 7 µL polymer electrolyte composed of 0.5 mol/L of Lithium bis(trifluoromethanesulfonyl)imide (LiTFSI) in methyl methacrylate -polyethylene glycol (MMA-PEG) was drop casted onto the surface of the ring electrodes. The electrodes were then dried at 70 °C for 18 h to obtain a homogeneous polymer thin film. Finally, the ring micro battery was assembled in an argon filled glovebox (Jacomex) with <0.5 ppm H_2_O and <0.5 ppm O_2_ atmosphere. Figure 4a shows an optical image of the Laser patterned electrode and Figure 4b the 0.75 cm^2^ ring micro battery powering an LED in the orange, respectively.

## 5. Measurements and Tests

Figure 5a shows the galvanostatic cycling of the all-solid-state micro battery LTO/Polymer/LNMO in the potential window of 1 V−3.5 V at C/10 for the first, and 10th cycles. The micro battery shows an operating voltage at 2.55 V. This is consistent with the reversible intercalation of the Li^+^ for LTO and LNMO occurring at 1.55 V and 4.7 V vs. Li/Li^+^, respectively. The micro battery shows in the first reversible cycle a charge and discharge areal capacities of 1.22 mAh·cm^−2^ and 1.196 mAh·cm^−2^, respectively. The coulombic efficiency for the first reversible cycle corresponds to 96.17%. The discharge area capacities for the 2nd, 5th, and the 10th cycles are 1.183 mAh·cm^−2^, 1.16 mAh·cm^−2^, and 1.181 mAh·cm^−2^, respectively. The corresponding coulombic efficiencies of these cycles were 92.77%, 93.43%, and 92.91%, respectively. Regarding the cycling performance, the LTO/polymer/LNMO micro battery has been assessed at fast kinetics for 30 cycles, Figure 5b. The micro battery delivers 73.5 µAh at 6C, 47 µAh·cm^−2^ at 12C and 32 µAh·cm^−2^ at 20C with a remarkable stability. The main characteristics of the micro-battery are summarized in Table 2.

## 6. Discussions and Conclusions

In this paper, we demonstrated the first attempt of encapsulating a flexible battery into a contact lens. Preliminary measurements have shown the battery capability to power an external LED. Of course, at this development stage, the energy is sufficient to power the future full system that will include an ASIC with a RF transceiver during a long time. Powering only the RF transmitter (all the other blocks being off) necessitates drawing 590 µA from the battery. The battery presented can deliver this value at 20C, for a duration of 3 min only, while transmitting the data will require shorter durations (dozens of ms at most). These figures are encouraging since they demonstrate that the battery can power the ASIC in full working mode for a short duration, longer duration would be achieved using power management techniques. The main source of power consumption being the RF transmitter, efforts should be dedicated to reduce its power. This could be done, since the data link is done over a very short distance. The battery capacity will be certainly increased in the future thanks to material developments such as improvement of adhesion and the ionic conductivity of the electrolyte, and creating micro structured electrodes to sustain the mechanical stress of the micro battery and increasing the surface area, which will enhance the capacity. Besides, in practice, a double layer could be stacked easily by reducing the substrate layer thickness, as well as using the pupil area provided transparent electrodes could be implemented thanks, for instance, to graphene technology. However, in general, except in the case of disposable CLs, the battery should be seen rather as a required complementary element for the harvesting function. Disposable options have to be considered in the case where the electronics is simplified as well as the data transfer protocol and that no ASICs are involved in the CL.

## Figures and Tables

**Figure 1 sensors-19-02062-f001:**
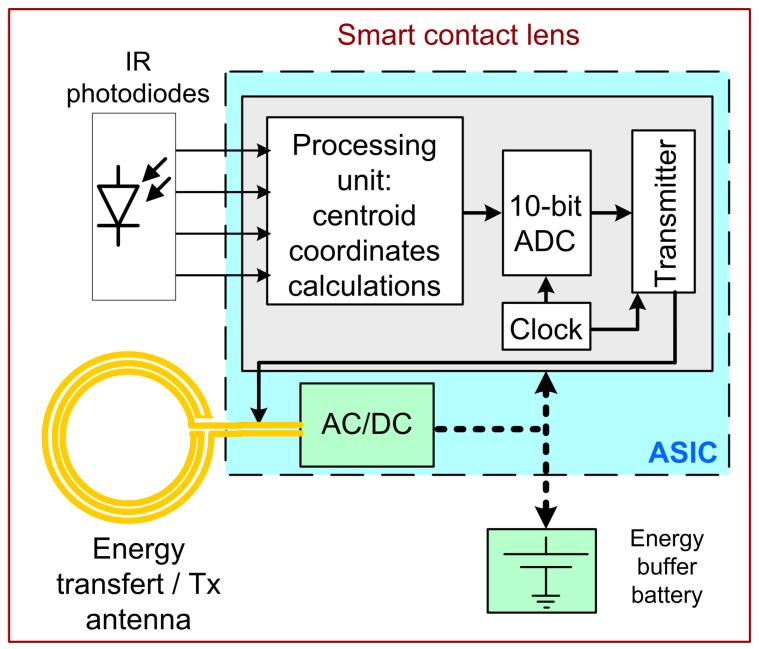
Block diagram of the smart contact lens including a simplified description of the ASIC functions.

**Figure 2 sensors-19-02062-f002:**
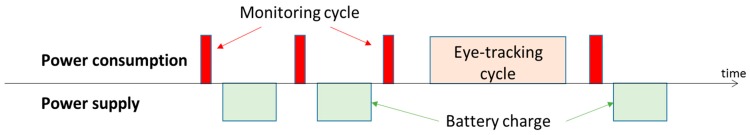
Cycles of use and the resulting cycle of charge and consumption.

**Figure 3 sensors-19-02062-f003:**
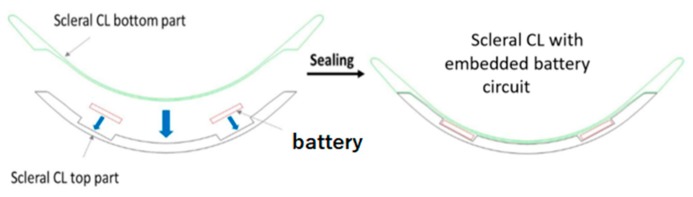
Principle of the battery encapsulation in a scleral contact lens (from LCS).

**Figure 4 sensors-19-02062-f004:**
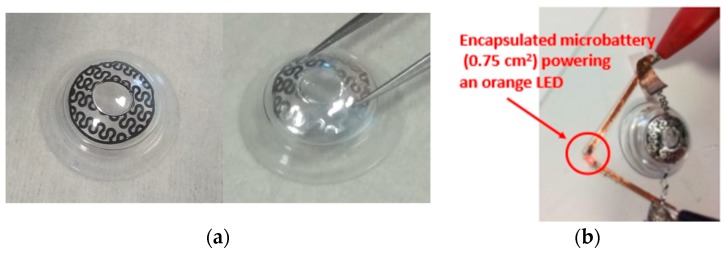
Flexible laser patterned LNMO electrode (**a**), prototype 0.75 cm^2^ ring micro-battery powering a LED (KPG-1608SEKC-T from Kingbright™ emitting @ 610 nm) (**b**).

**Figure 5 sensors-19-02062-f005:**
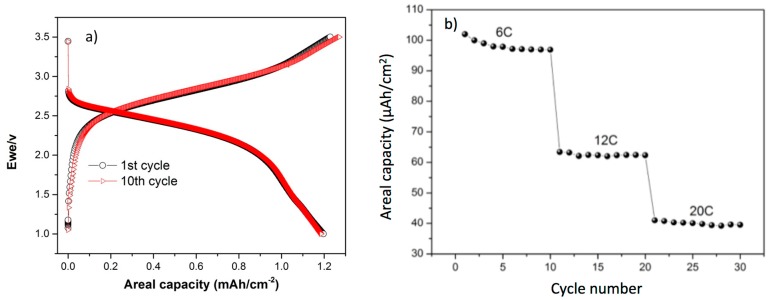
Galvanostatic charge/discharge profiles of LTO/Polymer/LNMO micro battery (**a**); discharge capacity of the micro battery at multi C-rates (**b**).

**Table 1 sensors-19-02062-t001:** Smart contact lens energy budget.

Estimated harvested RF power * (mW)	≈2
Simulated consumed ASIC power (mW)	
Processing unit	0.043
Analog to Digital Converter	0.015
Clock oscillator	0.1
RF transmitter	0.7

* for the harvesting antenna, the key parameter is the power transfer efficiency (PTE) which depends on the frequency and physical mechanisms (e.g., magnetic coupling). A value between 15%–20% is generally considered. For a contact lens, the Specific Absorption Rate related to RF energy absorbed by human tissues and temperature determines the maximum energy of the emitting source with respect to current regulations (FCC, ICNIRP and IEEE). We consider few tens of mW as representative.

**Table 2 sensors-19-02062-t002:** Battery characteristics.

**Discharge Rate**	C/10	6C	12C	20C
**Discharge Time (min)**	600	10	5	3
**Areal Capacity (µAh·cm^−2^)**	1200	98	63	43
**Capacity (µAh)**	900	73.5	47	32
**Delivered Current (µA)**	90	441	564	640
**Energy Density (µWh)**	2295	187	120	81
**Power Density (µW)**	229.5	1125	1438	1632

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
