# Peer review of "Flexible Micro-Battery for Powering Smart Contact Lens"

_sensors, 2019, doi:10.3390/s19092062_

Reviewer 1 Report

Enjoyed the technical merits of this paper. Have several concerns throughout the manuscript that must be addressed to ease reading. All will be detailed below.

Introduction

-lines 26-29- several typos, including in a second last part plus more...some future extends....

Section 2 Battery requirements

-lines 50-54 more typos

-lines 58-61 argument on energy harvesters remaining small, talks about lacrimation but doesn't expound on the concept, add more material to explain adequately 

-lines 62-3 more typos, monitoring regime "during which"

-lines 74-5 less has that harvested....

-lines 77-80 multiple typos including require a RF transceivers, plus

-lines 90-92 re-word syntax poor

-line 89 suggest expounding on work in REF 89 to include more material and add details

Section 3 Principle of encapsulation 

-lines 99-100 re-word poor syntax

-lines 102 electronics dimensions, not correct - typo

-lines 108-110 re-word poor syntax

Section 4 Micro-battery (spelling not consistency throughout doc, fix)

-lines 113-4, typo 

-line 115 innovative micro-battery suggest

-lines 120-139 discuss procedure in detail but entitle this section principle?, needs to be addressed and re-worked

Section 5 Measurements and tests

-line 151 no "a" before remarkable

Section 6 Discussion 

-lines 161-3 several typos

-lines 164-6 re-word poor syntax

-lines 168-172 expand on this section (material developments)

-add concluding statement in review of technical work

Figures:

-Fig 1 in legend the harvesting concept with buffering reversed, revise

-Fig 2 have color codes for different 'blocks' but there is no explanation of color or a figure legend to explain color

-Fig 3 removal scleral from all the phrases in the Fig, redundant 

-Fig 4 magnified view in middle lower left, blurry, re-take

-Fig 5 separate colored lines or change girth so the lines are differentiable 

Author Response

Reviewer 1

Introduction

-lines 26-29- several typos, including in a second last part plus more...some future extends.... Done

Section 2 Battery requirements

-lines 50-54 more typos Done

-lines 58-61 argument on energy harvesters remaining small, talks about lacrimation but doesn't expound on the concept, add more material to explain adequately (we added some figures in the text)

-lines 62-3 more typos, monitoring regime "during which" Done

-lines 74-5 less has that harvested.... Done

-lines 77-80 multiple typos including require a RF transceivers, plus Done

-lines 90-92 re-word syntax poor Done

-line 89 suggest expounding on work in REF 89 to include more material and add details. We detail and add a reference [16].

Section 3 Principle of encapsulation 

-lines 99-100 re-word poor syntax Done

-lines 102 electronics dimensions, not correct – typo Done

-lines 108-110 re-word poor syntax Done

Section 4 Micro-battery (spelling not consistency throughout doc, fix). Done micro battery.

-lines 113-4, typo Done

-line 115 innovative micro-battery suggest Done

-lines 120-139 discuss procedure in detail but entitle this section principle?, needs to be addressed and re-worked Done

Section 5 Measurements and tests

-line 151 no "a" before remarkable Done

Section 6 Discussion 

-lines 161-3 several typos Done

-lines 164-6 re-word poor syntax Done

-lines 168-172 expand on this section (material developments) Done

-add concluding statement in review of technical work. Proposed

Figures:

-Fig 1 in legend the harvesting concept with buffering reversed, revise. Done

-Fig 2 have color codes for different 'blocks' but there is no explanation of color or a figure legend to explain color . Since the lines are overlapping we reduce the number of lines to show only the first and the 10th cycles with symbols to show the stability of the micro battery (new figure).

-Fig 3 removal scleral from all the phrases in the Fig, redundant.

-Fig 4 magnified view in middle lower left, blurry, re-take (we simplify the picture)

-Fig 5 separate colored lines or change girth so the lines are differentiable  Done see above

Reviewer 2 Report

This manuscript proposed a high-level smart contact lens design with on-lens battery, but the authors should provide more detail description about system design and consideration.

1. The authors should provide system block diagram such as that in reference [2], the overall system desgin consideration should be mentioned in introduction at least.

2. The authors seems to skip much key information and the paragraphs in the manuscript do not link to each very well.

3. In section 2, the authors discuss the requirements of on-lens battery, it should be better to describe the power transfer flow. For example, tx power and antenna efficiency.

4. The power consumption of ASIC is not very clear, the authors should provide system diagram of ASIC and clearly describe the power consumption estimation.

5. In section 4, the autors activate an orange LED but without I-V information of LED. it is the fundamental for power harvesting design.

6. The activation of orange LED is not very clear, is it activated by battery ? or it is activated by external power supply when the on-lens battery is charged ?

Author Response

Reviewer 2

1. The authors should provide system block diagram such as that in reference [2], the overall system desgin consideration should be mentioned in introduction at least. We added a block diagram

2. The authors seems to skip much key information and the paragraphs in the manuscript do not link to each very well. We tried to improve by reorganizing the sections.

3. In section 2, the authors discuss the requirements of on-lens battery, it should be better to describe the power transfer flow. For example, tx power and antenna efficiency. RF-FET is given in the figure caption, figures have been added consequently.

4. The power consumption of ASIC is not very clear, the authors should provide system diagram of ASIC and clearly describe the power consumption estimation. (Done see above)

5. In section 4, the autors activate an orange LED but without I-V information of LED. it is the fundamental for power harvesting design. LED characteristics and supplier have been added in the figure caption 4.

6. The activation of orange LED is not very clear, is it activated by battery ? or it is activated by external power supply when the on-lens battery is charged ? activated by the battery clarified in the text.

Reviewer 3 Report

This paper investigates the application of flexible micro-battery for powering smart contact lens. The findings and conclusions are in the scopes of Sensors, but the authors need to properly address the following concerns before leading to a publication.

1.      The reviewer feels that the title of the paper is inappropriate. Given the study mainly focuses on the energy solution to an existing technology (smart contact lens), the title should be more specific rather than broadly mentioning the lens.

2.      Abstract is too brief. It would be much helpful to readers if the authors can introduce the main findings in the study.

3.      Introduction needs to be much improved. On the one hand, the reviewer doesn’t see the necessity of fig. 1. If the authors mean to compare generated and consumed power in contact lens, more information can be delivered such as comparing different energy harvesting techniques. On the other hand, the authors have not provided adequate information in the field. For example, … (e.g., optical [7], mechanical [8]…), the authors need to review and discuss more studies in the literature such as,

·         Shi et al. (2019). Nano Energy. 57, 851-871.

·         Radgolchin and Moeenfard (2018). SMS. 27, 025015.

·         Jiao et al. (2017). SMS 26, 085045.

In addition, please be specific what 1st, 2nd and 3rd parts refer to.

4.      Seems to the reviewer, Sec 2 is a little confusing. The authors mention two requirements for battery in contact lens, and it may be easier to follow if they introduce what the existing devices are and how the presented technique is better and why? Following this point, the reviewer tends to think Sec 3 can be combined with sec 2, to make the comparison more solid.

5.      Sec 4 is the core in this study and thus, it could be better if the authors can separate it into two parts, such as fabrication and testing setup. Again, it’s a little hard to follow.

6.      Efficiency is, in general, a key issue for energy harvesting techniques. The authors need to compare the reported method with studies in the literature to exhibit the advantages and why. Also, it would be much helpful if the authors can provide certain parametric studies in Sec 5.

7.      Some minor suggestions: 1) the figures are not proper such as Fig. 3 is too blurred. The zoom-in figure in 4(b) is not clear, and different types of lines need to used in 5(a).

 Author Response

Reviewer 3

1.      The reviewer feels that the title of the paper is inappropriate. Given the study mainly focuses on the energy solution to an existing technology (smart contact lens), the title should be more specific rather than broadly mentioning the lens. We changed the title.

2.      Abstract is too brief. It would be much helpful to readers if the authors can introduce the main findings in the study. We developped the abstract.

3.      Introduction needs to be much improved. On the one hand, the reviewer doesn’t see the necessity of fig. 1. If the authors mean to compare generated and consumed power in contact lens, more information can be delivered such as comparing different energy harvesting techniques. On the other hand, the authors have not provided adequate information in the field. For example, … (e.g., optical [7], mechanical [8]…), the authors need to review and discuss more studies in the literature such as, We canceled Fig1 as suggested and added a new general reference considering that harvesting sources are multiple (even if most of them are not compatible with a contact lens technology).

·         Shi et al. (2019). Nano Energy. 57, 851-871.

·         Radgolchin and Moeenfard (2018). SMS. 27, 025015.

·         Jiao et al. (2017). SMS 26, 085045.

In addition, please be specific what 1st, 2nd and 3rd parts refer to. Done

4.      Seems to the reviewer, Sec 2 is a little confusing. The authors mention two requirements for battery in contact lens, and it may be easier to follow if they introduce what the existing devices are and how the presented technique is better and why? Following this point, the reviewer tends to think Sec 3 can be combined with sec 2, to make the comparison more solid. We tried to clarify

5.      Sec 4 is the core in this study and thus, it could be better if the authors can separate it into two parts, such as fabrication and testing setup. Again, it’s a little hard to follow. Done

6.      Efficiency is, in general, a key issue for energy harvesting techniques. The authors need to compare the reported method with studies in the literature to exhibit the advantages and why. Also, it would be much helpful if the authors can provide certain parametric studies in Sec 5. The paper did not deal with the harvesting (many review papers address this point already) but with on-lens battery, even if harvesting is considered as a necessary complementary option.

7.      Some minor suggestions: 1) the figures are not proper such as Fig. 3 is too blurred. The zoom-in figure in 4(b) is not clear, and different types of lines need to used in 5(a). Done

Round  2

Reviewer 2 Report

It will be great to do some measurement with discrete components, such as overall measurement including ASIC and battery.

Author Response

Measurements of a full electronic system (even built from off-the-self electronic devices)  powered by the battery would be interesting but are the object of the next step. This is why the capacity of the battery to power electronic devices is simply demonstrated by powering the LED. The main goal being to show that we have a battery that can do the job. The ASIC is currently being designed and the simulations provided enough results to dimension the battery. Putting everything together demands a lot of work that is underway.

 We hope that you will feel the presented work worthy of publication.

English language and style are fine/minor spell check have been performed. Corrections appear in blue in the uploaded version of the manuscript.                  

Reviewer 3 Report

The authors have properly addressed my concerns on the manuscript and thus, I suggest accepting it for publication. 

Author Response

The manuscript has been corrected for typos and English .

Corrections and rephrasing appear in blue in the uploaded word document.